# Neural response to sad autobiographical recall and sad music listening post recall reveals distinct brain activation in alpha and gamma bands

**Ashish Gupta[1], Braj Bhushan [2], Laxmidhar Behera [1,3] ***

**1** Department of Electrical Engineering, Indian Institute of Technology, Kanpur, India, **2** Department of Humanities and Social Sciences, Indian Institute of Technology, Kanpur, India, **3** School of Computing and Electrical Engineering, Indian Institute of Technology, Mandi, India

* lbehera@iitk.ac.in

**Data Availability Statement:** All data files are available from the Open Science Framework database (DOI 10.17605/OSF.IO/3U2GT). The files

## Abstract

Although apparently paradoxical, sad music has been effective in coping with sad life experiences. The underpinning brain neural correlates of this are not well explored. We performed Electroencephalography (EEG) source-level analysis for the brain during a sad autobiographical recall (SAR) and upon exposure to sad music. We specifically investigated the Cingulate cortex complex and Parahippocampus (PHC) regions, areas prominently involved in emotion and memory processing. Results show enhanced alpha band lag phase-synchronization in the brain during sad music listening, especially within and between the Posterior cingulate cortex (PCC) and (PHC) compared to SAR. This enhancement was lateralized for alpha1 and alpha2 bands in the left and right hemispheres, respectively. We also observed a significant increase in alpha2 brain current source density (CSD) during sad music listening compared to SAR and baseline resting state in the region of interest (ROI). Brain during SAR condition had enhanced right hemisphere lateralized functional connectivity and CSD in gamma band compared to sad music listening and baseline resting state. Our findings show that the brain during the SAR state had enhanced gamma-band activity, signifying increased content binding capacity. At the same time, the brain is associated with an enhanced alpha band activity while sad music listening, signifying increased content-specific information processing. Thus, the results suggest that the brain's neural correlates during sad music listening are distinct from the SAR state as well as the baseline resting state and facilitate enhanced content-specific information processing potentially through three-channel neural pathways—(1) by enhancing the network connectivity in the region of interest (ROI), (2) by enhancing local cortical integration of areas in ROI, and (3) by enhancing sustained attention. We argue that enhanced content-specific information processing possibly supports the positive experience during sad music listening post a sad experience in a healthy population. Finally, we propose that sadness has two different characteristics under SAR state and sad music listening.

can be accessed at https://doi.org/10.17605/OSF.
IO/3U2GT.

**Funding:** The authors received no specific funding
for this work.

**Competing interests:** The authors have declared
that no competing interests exist.

## Introduction

Music and the human brain are strongly connected from an evolutionary perspective [1]. Humans have used happy and sad music to boost their emotions [2]. They generally prefer happy music [3] and endeavour to minimize sadness in their life. However, paradoxically, they have a great longing for sad music [4], especially during negative situations ranging from daily sorrows [5] to difficulties in relationships, dramatic occurrences like the loss of loved ones [5], etc. Sad music, in general, has been shown to induce genuine sadness [6] and affect cognitive abilities [6] like memory and judgement [7] adversely. Nevertheless, studies have shown that sad music also brings about a pleasurable experience [8] ranging from a sense of solace to a state of profound beauty in addition to sadness [4], a blend of positive and negative emotions [5], a feeling of "being moved" [5]. Behavioural studies found that sadness elicited during listening to sad music was qualitatively different [9] and significantly more pleasant than the real-life sadness [7]. This mixed experience has led to different views regarding the nature of sadness induced through sad music. Emotivists believe that sad music induces genuine emotions [6, 10]. On the other hand, cognitivism proposes that the emotions elicited through music are purely aesthetic and have little to do with real emotions [8, 11]. While other authors argue that the emotional responses to sad music are genuine but are different from those during real-life negative situations [6, 12].

Nevertheless, the beneficial outcomes of listening to sad music in coping with unfavourable circumstances are well documented [3, 5, 13]. Psychologically healthy adolescents [14] and young adults have used sad music for solace [3], consolation [15], comforting [4] and emotional coping [3]. Several self-regulatory goals in the domains of cognitive, social, retrieving memories, friend, distraction, mood enhancement, and re-experience affect [3, 16] have consistently been observed upon listening to sad music during adverse circumstances. Results show re-experiencing effect as the ability to get in touch with one's feelings and express them was the most prominent function of sad music [3]. A review on the usage of sad music concluded that although psychologically healthy individuals tend to benefit from sad music listening, the authors cautioned that the usage of sad music among individuals with a history of depression, PTSD, etc., could be maladaptive [14].

Sad Autobiographical recall (SAR) is among the most effective methods to induce and re-experience a realistic mood in a laboratory setting [17, 18]. Studies have shown the effectiveness of sad music in elevating mood in the participants primed into negative mood [19] through SAR [18]. Although the brain during sad music listening is characterized by several self-regulatory goals like (1) regulation of negative emotions, (2) enhanced emotion expression, (3) enhanced memories retrieval and (4) being more pleasant than real-life sadness. However, it is unclear whether the self-regulatory goals achieved during sad music listening are just artefacts of the aesthetic feature of music or if there is potential brain neural processing in the relevant regions supporting them. To our best knowledge, no study has directly investigated (1) the neural correlates of listening to sad music [8, 12] post adverse situation, and (2) differences in the neural correlates of listening to sad music as compared to the adverse circumstance [8, 12] in the emotion and memory processing area. Thus, to study the brain during sad music listening in an adverse circumstance, we used a standard mood induction procedure, SAR [17], to relive a sad real-life event and experience its emotions. An eastern classical composition sung by a trained Indian musician was used as the sad stimulus post-SAR. The stimulus maintains the cultural salience of the music for the participants.

Reviews on the neural basis for emotion and memory have found two limbic system structures prominently underpinning emotion and memory process [20, 21]. Anterior cingulate cortex (ACC) along with amygdala and orbital frontal cortex in emotion processing [20].

Posterior cingulate cortex along with hippocampus and parahippocampus in-memory processing [20]. ACC is involved in emotion assessment [22], regulation [23], learning [22], and stores reward/value specific information of the stimulus [24]. Studies have shown that ACC is directly and linearly related to the subjective experience of pleasantness and unpleasantness of the stimulus [25–27]. Midcingulate cortex (MCC) synthesizes reward/value related information from ACC, and spatial information from PCC [28, 29] for action-outcome learning [21]. It is also found to be involved in emotional appraisal and reappraisal [22, 23]. In addition to spatial processing of space and action, PCC [28, 29] has significant connections to the PHC area. Thus plays a vital role in the memory process [28, 29] example episodic memory formation [30, 31]. The precision of the memory recall (spatial context) has been positively correlated with the strength of the connections between PCC and PHC [32, 33]. Indeed interconnectivity dynamics of the cingulate cortex complex and PHC play an essential role in emotion and memory processing [21, 34, 35]. Studies have also shown that these associated areas get activated during SAR, viz. anterior and posterior cingulate [36], hippocampus [37], and PHC [36, 37]. Moreover, sad music has been shown to be effective in modulating anterior cingulate cortex [38], posterior cingulate cortex [39], PHC [38], etc.

The Valance lateralization hypothesis postulates that the left hemisphere is specialized in processing positive emotions, and the right hemisphere is specialized in processing negative emotions. A recent review favoured the right hemisphere dominance hypothesis that emotions, irrespective of the valance, are primarily processed in the right hemisphere [40]. Lateralization studies for memories have also found right hemisphere dominance during negative autobiography memory retrieval [41–44]. Since our experiment involves negative valanced emotions and memories, we expect the results to have right hemisphere dominance.

EEG research of episodic memory encoding and retrieval suggests the involvement of low-frequency range oscillation (alpha/beta) event-related desynchronization (ERD) in processing the contents of episodes and high range oscillation (gamma) event-related synchronization (ERS) in binding them [45, 46]. Gamma oscillation in the cortex helps facilitate perceptual features binding while in the hippocampus and related area helps bind perceptual and contextual information into an episode [47]. The finding from the music EEG study shows an increase in ERS in the frequency band of theta, alpha and beta upon listening to music [48, 49]. Brain under music listening is characterized by a neurophysiological and psychological state of increased internal attention, alertness and diminished external attention [48, 49]. This state is similar to that under meditation, imagination, creative thinking, etc [48, 49]. Enhanced alertness positively correlates with the activity of the dorsal anterior cingulate cortex, particularly in the alpha2 band [50, 51]. Several studies have performed functional connectivity analysis upon listening to music. Results show a positive correlation between alpha band phase synchronization and neuronal excitability of task-relevant areas [52]. Enhanced alpha band phase synchronization was correlated with enhanced emotional and perception [52–54] processing for both pleasant and unpleasant music.

In summary, we investigated the brain neural correlates while reliving a sad real-life instance and its alteration upon passive listening to a sad composition. We studied EEG neural correlates of the brain at the source level, particularly in areas related to emotion and memory (cingulate cortex complex and PHC). We expected that brain neural correlates during sad music listening would be unique and distinct from that during the negative situation. Our first investigating tool was brain functional connectivity measured through lag-phase synchronization. We hypothesized that (1) enhanced functional connectivity, especially between PCC and PHC, during sad music listening. Our following investigating tool was brain region activation estimated through current source density (CSD). Furthermore, we hypothesized that (2) SAR

state to be characterized with ERD in the alpha/beta band and ERS in the gamma band, and (3) an increased ERS in low-frequency range oscillation during the sad musical intervention.

## Methods

### Participants

Twenty right-handed participants with no musical training were enrolled from a technology institute. The participants were recruited through an electronic announcement. Their age ranged between 19–29 years (mean age = 22.14 years, SD = 3.68). Only male participants were considered in the study, as the brain networks used by males and females for processing music stimuli are shown to have differences [55]. Any neurological or hearing disorders and usage of psychotic drugs in recent times were the exclusion criteria. Participants were also excluded if their response was yes to the following questions "During the past month, have you often been bothered by feeling down, depressed, or hopeless?" or "During the past month, have you often been bothered by little interest or pleasure in doing things?" [56]. The study was approved by the Institutional Ethics Committee (IEC) of the Indian Institute of Technology Kanpur (protocol number: IITK/IEC/2019–20/I/18) for research involving human subjects of the Institute. All experiments were performed in accordance with the relevant guidelines and regulations. All participants duly filled informed consent form in writing before the conduction of the experiment.

### Stimulus selection

The musical stimulus selection was made based on the rating of the panel of five musically expert judges [57]. Each panel member was asked to bring three compositions of Indian classical music (without lyrics) that convey sadness. All the panel members rated the resulting musical experts for the emotion expressed using five discrete emotion scales (happiness, sadness, anger, fear, and tenderness) as used in an earlier study [7]. Emotion ratings were done on a five-point scale. The excerpt, which predominantly expressed sadness and rated highest, was selected for the experiment. The selected musical piece was a Mishra jogiya raag and is known for invoking sad emotions [58]. It was an "alap in akar" of 8 min, 44 sec duration (performed by Nusrat Fateh Ali Khan).

### Experimental procedure

All the participants were administered Kessler Psychological Distress Scale to screen out any participant prone to depression or anxiety disorder [56]etc. Participants with scores greater than 20 were screened out [56]. This was to prevent potentially maladaptively usage of sad musical stimulus during sad emotion regulation by participants with the tendency for depression etc [59, 60]. Participants sat in an adjustable chair in front of a table. A pillow was used in the back to release any tension during the experiment. The experimental sessions consisted of three conditions—(1) baseline resting-state condition for 9 min, (2) sad autobiographical recall condition for 9 min (participants recalled an event from their life that made them sad), and (3) sad music condition (passive hearing of a segment of the selected musical stimulus). All the participants were instructed to sit calmly with their gaze fixed on the cross printed centrally on a blank paper kept over the table during baseline and sad music listening conditions. The cross sheet was replaced with a writing pad upon the table during SAR. Participants were told to recall and write about a negative real-life event in which they experienced sadness like feelings of loss, loneliness, misunderstanding, heartbreak, betrayal, loss of a loved one, etc [5]. Participants were encouraged to relive and report the event as vividly and detailed as possible [7, 17]

in the writing pad. They were asked to support their elbow while writing in order to reduce their hand movements. All the participants were also requested to sit as still as possible while simultaneously performing the task as naturally as possible. This was to minimize artifacts from eyes, head and body movement. We selected autobiographical memory recall for sad mood induction since it is one of the best methods for effectively inducing real-life emotion. The covert recall might also foster better capturing of the moment-by-moment brain dynamics as the sad event unfolds mentally. The entire EEG experiment was performed inside a sound-proof chamber in dim light. The musical stimulus was given through a loudspeaker placed symmetrically around the participants.

## EEG recording and preprocessing

We used a g.HIamp bio-signal amplifier (Guger Technologies, OG, Graz, Austria) to capture the EEG from the participants. EEG was sampled at a frequency of 512 Hz with less than 5 Kohms impedance level at 32 scalp locations (as per the International 10–20 system). We applied a band-pass filter of 0.01–100 Hz. We used the left earlobe as a reference electrode. We also recorded EEG at 4 EOG positions (On top and bottom of the right eye and either eye's outer canthus location) to eliminate any artifacts due to eye blink. EEG data were down-sampled to 256 Hz. A 0.5 Hz high pass filter was applied to eliminate any DC drift. We visually checked the EEG data for any eye, muscle, or electrode movement artifacts. Subsequently, we marked the bad electrodes and interpolated them. EEG data were then re-referenced to average reference. Independent component analysis (ICA) and SASICA (Semi-Automatic Selection of Independent Component Analysis) were used to further remove the artifacts due to eye and muscle movements after adjusting the rank. ICA and SASICA are implemented in the EEGLAB toolbox and are effective in eliminating the artifacts from eyes and muscle movements [61, 62]. ICA-based algorithms have been quite successful in cleaning the EEG data of eyes and muscles [63]. SASICA helps in the selection of the eye and muscle artifacts based upon several algorithms in addition to SASICA, like FASTER (Fully Automated Statistical Thresholding for EEG artifacts Rejection), ADJUST (Automatic EEG artifacts Detection based on the Joint Use of Spatial and Temporal features) [64]. This makes SASICA a promising tool for artifacts corrections. EEG data of 8 min and 44 sec long was analyzed for all three conditions. One participant's data was further omitted from the study due to high artifact. All our further analysis was done on average referenced EEG data. We performed all the investigations in the ROI compromising of emotion and memory regions.

## EEG source analysis

Exact Low Resolution Electromagnetic Tomography (eLoreta) is a weighted minimal norm inverse method and performs a three-dimensional, discrete and linear source localization. It is uniquely equipped with the ability to exactly localize a test point source [65]. Source localization and analysis of 32 channel EEG data were conducted through eLoreta. Intracortical current source density (CSD) was calculated at 6239 voxels, sampled at 5 mm resolution. CSD was calculated as a linear weighted sum of potentials at the scalp through eLoreta software. The following frequency bands were analysed: delta (1–4 Hz); theta (4.5–8 Hz); alpha1 (8.5–10 Hz); alpha2 (10.5–12 Hz); beta1 (12.5–18 Hz); beta2 (18.5–21 Hz); beta3 (21.5–30 Hz); gamma (35–44Hz). Brain activity was analysed across 689 voxels related to CC, PCC, ACC, and PHC. 689 Voxels are defined according to the standard Montreal Neurological Institute (MNI) template in our analysis area.

## Source connectivity analysis

Brain regions coordinate with one another to accomplish any cognitive or perceptual tasks. Functional connectivity between two brain regions during the task gives an index for how well these regions synchronized to accomplish the task [66]. Information flow between two regions is directly proportional to functional connectivity between them. Thus, functional connectivity is a good instrument to quantify how well the regions are connected in a particular task/condition. We selected functional connectivity as our first investigating tool. We chose lag phase synchronization for it measures non-linear functional connectivity independent of power fluctuation, any instantaneous zero lagged component, and volume conduction and is resistant to non-physiological artifact, particularly low spatial resolution [65]. Brain connectivity was computed by measuring lagged phase synchronization through eLoreta at the source level between all pairs of 27 ROI resulting in 729 connections. 27 ROI were defined according to the standard Montreal Neurological Institute (MNI) template and Brodmann areas (BA) in our area of analysis (cingulate cortex complex and PHC). 13 ROI were present on either side of the hemisphere and one in the central region (for the complete co-ordinates of ROI, see S1 Table).

## Subjective assessment

Subjective assessment of the sad autobiographical memory recall was done on vividness and reliving through a five-point scale (1 = very low, 5 = very high) [67]. Participants also rated their mood (between happy and sad) on an 11-point Likert scale across the three conditions. After the EEG experiment, subjective assessment for self-regulatory goals achieved during music listening was also taken via standard Self-Regulatory Goals Assessment [3] questionnaire. Two participants were excluded through the Kessler Psychological Distress Scale questionnaire's response. We also excluded the data of two participants who had previous exposure to the musical stimulus. All the participants were also made to see positive Velten-type statements post-experiment to neutralize any sadness-inducing effects of the experiment [68]. The participants were also asked to rate their experience on the effectiveness of sad musical stimulus for coping in the SAR condition on an 11-point bi-direction scale ranging from (-5 to +5).

## Statistical analysis

A two-tailed t-test with $\alpha$- value of 0.05 was used to compare the mean values and analyse subjective assessments. Repeated ANOVA was used to investigate simple and interaction effects between bands and conditions for the mean value of EEG data. Bonferroni correction was performed for multiple testing across conditions (Baseline, SAR, Music) and bands. Post hoc pairwise comparison revealed a significant difference only in alpha1, alpha2, and gamma bands which were used for further analysis. We used SPSS toolbox for statistical analysis. Statistical analysis of eLoreta source-level data at the voxels level is characterized by multiple testing problems. We used the non-parametric mapping method (SnPM) via randomization statistical tests (threshold set at 5% probability, Fisher's permutation method) as implemented in eLoreta statistical package to address the multiple correction problem for testing across all voxels, connections and bands [69]. eLoreta uses 5000 randomizations to determine critical probability threshold values with correction for multiple comparisons across voxels and frequencies. SnPm method in Loreta analysis has been validated in several earlier studies [70].

# Results

## Mean phase coherence analysis

We performed a preliminary study of mean coherence across 27 ROI to identify the frequency band where the effect of sad music intervention is maximum. A two-way repeated ANOVA with factor bands (8: delta, theta, alpha1, alpha2, beta1, beta2, beta3 and gamma) and conditions (3: baseline, SAR, and sad music listening) was administered. We found a significant interaction effect with a Greenhouse-Geisser correction between bands and conditions ($F_{3.785, 52.991} = 15.303$, $p < 0.001$). One-way repeated measure ANOVA was performed to study the simple effect of bands. Results showed a significant effect of bands on functional connectivity for theta ($F_{2,28} = 3.711$, $p = 0.037$), alpha1 ($F_{2,28} = 13.328$, $p<0.001$), alpha2 ($F_{2,28} = 22.710$, $p< 0.001$), beta2 ($F_{2,28} = 3.915$, $p = 0.032$), and gamma ($F_{2,28} = 7.402$, $p = 0.003$) bands. Post hoc pairwise comparison with Bonferroni correction revealed a significant difference only in alpha1, alpha2, and gamma bands. Table 1 shows the mean value of Mean lag-phase synchronization for the whole brain across different conditions (sad music, SAR, baseline) along with the p-value for the difference between the conditions. High synchronization implies better functional connectivity among the regions of interest. Findings showed a significant enhancement in the functional connectivity for the Music condition as compared to the SAR condition in the alpha1 band ($t = 4.41$, $df = 14$, $p<0.002$, effect size = 1.1387), in the alpha2 band ($t = 5.74$, $df = 14$, $p < 0.002$, effect size = 1.4827) and a significant reduction in gamma band ($t = -3.97$, $df = 14$, $p < 0.002$, effect size = -1.0273). Results also showed a significant reduction in the mean lag phase-synchronization of the brain during SAR conditions as compared to baseline in the alpha1 band ($t = -3.94$, $df = 14$, $p < 0.002$, effect size = -1.0183) and alpha2 band($t = -5.53$, $df = 14$, $p<0.002$, effect size = -1.4283) only. Results showed that alpha1, alpha2, and gamma-band were the most affected band and hence were selected for further analysis (see Table 1).

## Phase coherence analysis for sad music condition and SAR

We calculated lag-phase synchronization for 27*27 ROI pairs through eLoreta and observed significant changes between sad music listening and SAR condition in the alpha1, alpha2 and gamma bands. High coherence (lag-phase synchronization) implies high functional connectivity. Fig 1(a) shows several interconnections of the brain with significantly enhanced coherence in the alpha1 band during the sad music state with t-value ranging from 4.6339 to 5.2226 and effect size ranging from 1.1965 to 1.3485. Most of the connections were between PHC to PCC and PHC to ACC. Fig 1(b) depicts that most of the interconnections with enhanced coherence were located in the brain's left hemisphere compared to the right hemisphere. The difference was statistically significant ($t = 14.6426$, $df = 14$, $p<0.001$, effect size = 3.7807). Fig 1(c) shows

**Table 1. Mean phase coherence analysis.**

| Bands/ Condns. | Mean: Ms | Mean: SAR | Mean: BL | p-value: Ms Vs SAR | p-value: Ms Vs BL | p-value: SAR Vs BL |
|---|---|---|---|---|---|---|
| Theta | 0.025817149 | 0.03172679 | 0.020874407 | 0.118866655 | 0.1964835549 | 0.035629220 |
| Alpha1 | 0.120878241 | 0.046470902 | 0.122535169 | 0.000593192* | 0.906776823 | 0.001469557* |
| Alpha2 | 0.167790108 | 0.051265335 | 0.182173521 | 5.09E-05* | 0.478311389 | 7.39E-05* |
| Beta2 | 0.032824653 | 0.021487982 | 0.028102876 | 0.024355147 | 0.214273223 | 0.123957443 |
| Gamma | 0.015758674 | 0.026168399 | 0.014148663 | 0.001372256* | 0.624069884 | 0.01196064 |

Mean Lag-phase Synchronization comparison of the whole brain across Ms: sad music, SAR, and BL: baseline resting state conditions.

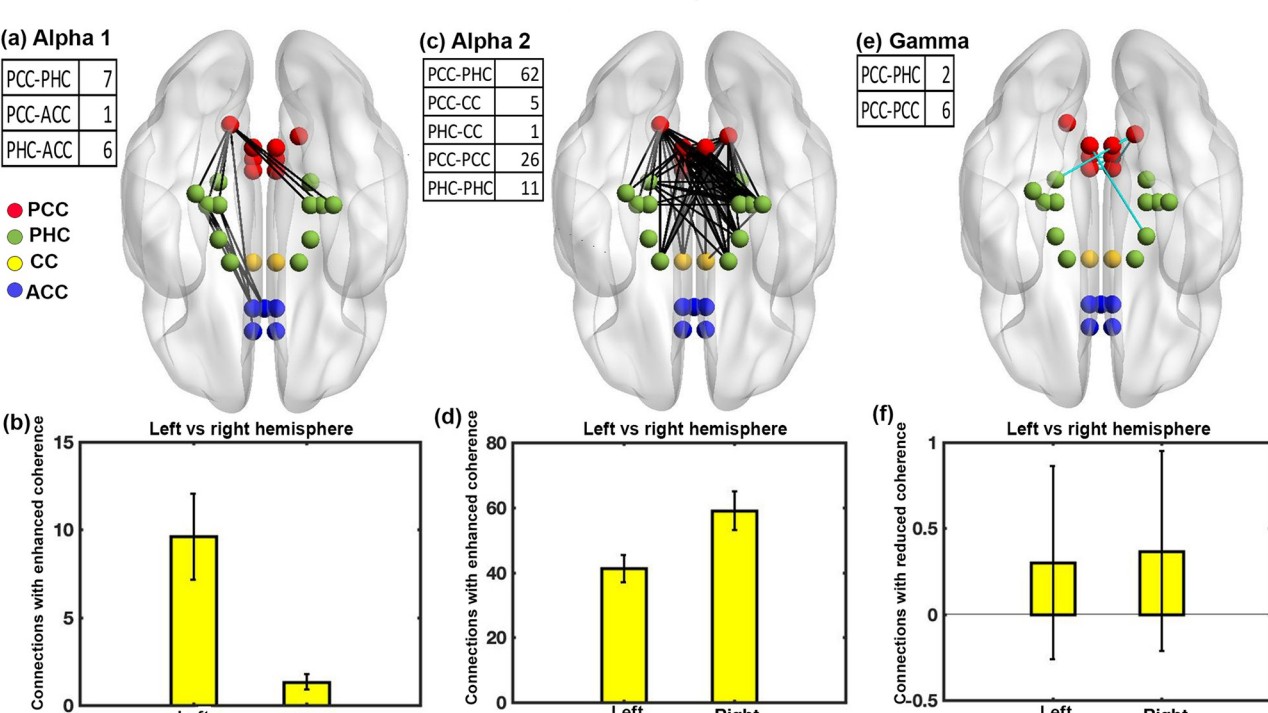

**Fig 1. Phase coherence comparison between sad music condition and SAR.** (a) Brain's connections (14) with a significant increase in the lag phase synchronization in alpha1 band, (b) Brain hemisphere lateralization with respect to inter-connections with increased coherence in alpha1 band,(c) Brain's connections (105) with a significant increase in the lag phase synchronization value in alpha2 band. (d) Brain hemisphere lateralization with respect to inter-connections with increased coherence in alpha2 band, (e) Brain's connections (8) with a significant decrease in the lag phase synchronization value in the gamma band (f) Brain hemisphere lateralization with respect to inter-connections with decrease coherence in gamma band (error bars = 1 SD).

several interconnections of the brain with significantly enhanced coherence in the alpha2 band during the sad music state with t-value ranging from 4.6322 to 7.5765 and effect size ranging from 1.1960 to 1.9562. Most of the interconnections were between PCC to PHC and within PCC and PHC. The interconnections with enhanced coherence were located mostly in the brain's right hemisphere compared to the left hemisphere. The difference was statistically significant (t = -25.8362, df = 14, p<0.001, effect size = -6.6709) as shown in Fig 1(d). Fig 1(e) shows several interconnections of the brain with significantly reduced coherence in the gamma band during the sad music state with t-value ranging from -4.638 to -5.8001 and effect size ranging from -1.1975 to -1.4976. Most of the interconnections were between PCC to PCC. We did not find any statistical difference in brain hemisphere lateralization for the interconnections with reduced coherence (t = -1, df = 14, p = 0.3343, effect size = -0.2582) as shown in Fig 1(f).

## Phase coherence analysis for SAR condition and baseline resting state

We calculated lag-phase synchronization for 27*27 ROI pairs through eLoreta and observed significant changes between the SAR condition and baseline resting state in the alpha1,alpha2 and gamma bands. Low coherence implies reduced functional connectivity. Fig 2(a) shows only one connection between PHC and cingulate cortex (CC) with significantly reduced

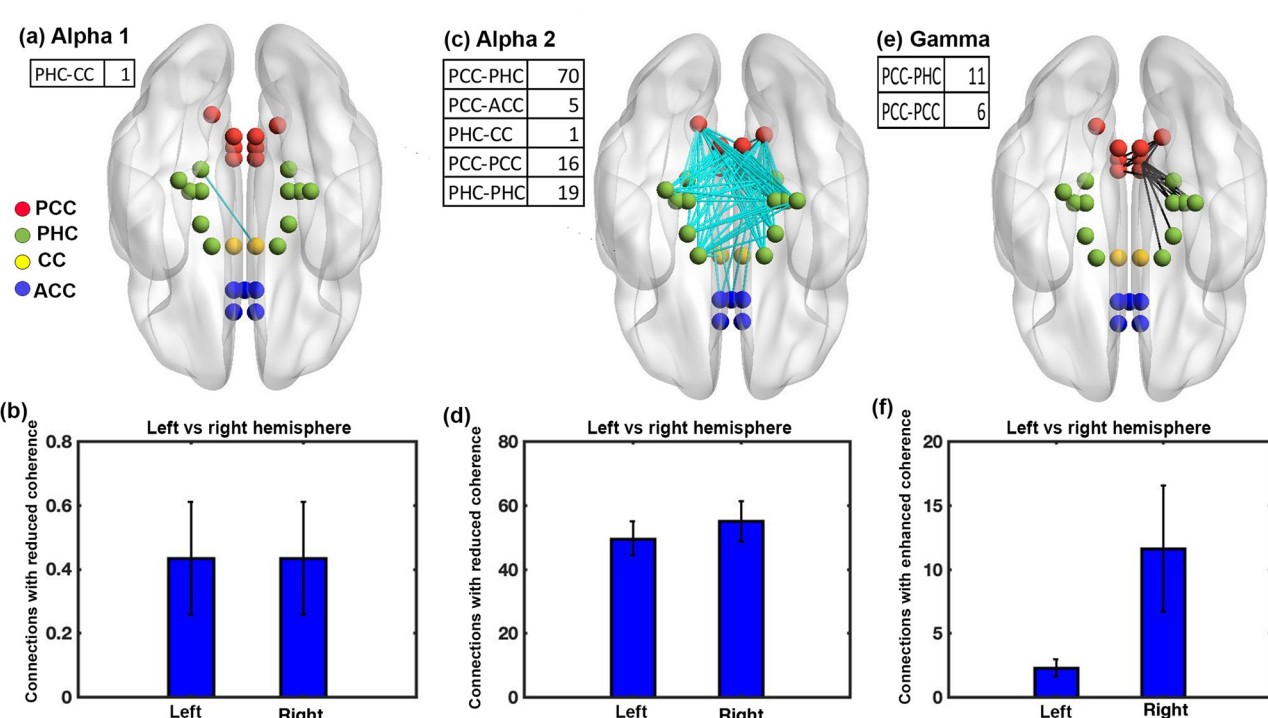

**Fig 2. Phase coherence comparison between SAR condition and baseline resting state.** (a) Brain's connections (1) with a significant decrease in the lag phase synchronization value in alpha1 band, (b) Brain hemisphere lateralization with respect to inter-connections with reduced coherence in alpha1 band,(c) Brain's connections (111) with a significant decrease in the lag phase synchronization value in alpha2 band. (d) Brain hemisphere lateralization with respect to inter-connections with reduced coherence in alpha2 band, (e) Brain's connections (17) with a significant increase in the lag phase synchronization value in gamma band (f) Brain hemisphere lateralization with respect to inter-connections with increase coherence in gamma band (error bars = 1 SD).

coherence in the alpha1 band during SAR condition (t = -4.6681, df = 14, p<0.167, effect size = -1.2053) as compared to resting state. We did not find any statistical difference in hemispheric brain lateralization for the interconnection with reduced coherence, as shown in Fig 2 (b). Fig 2(c) shows several interconnections of the brain with significantly reduced coherence in the alpha2 band during SAR condition with t-value ranging from -4.6652 to -6.7256 and effect size ranging from -1.2045 to -1.7365. Most of the interconnections were between PCC to PHC and within PCC and PHC. The interconnections with reduced coherence were located mostly in the brain's right hemisphere compared to the left hemisphere. The difference was statistically significant (t = -10.0111, df = 14, p<0.001, effect size = -2.5849) as shown in Fig 2 (d). Fig 2(e) shows several interconnections of the brain with significantly enhanced coherence in the gamma band during SAR state with t-value ranging from 4.6497 to 6.0957 and effect size ranging from 1.2005 to 1.5739. Most of the interconnections were between PCC to PHC and within PCC. The interconnections with enhanced coherence were located mostly in the brain's right hemisphere compared to the left hemisphere. The difference was statistically significant (t = -8.2110, df = 14, p< 0.001, effect size = -2.1201) as shown in Fig 2(f).

## Mean CSD analysis

Brain activity was estimated through CSD by eLoreta across 689 voxels in our analysis area (cingulate cortex complex and PHC) for alpah1, alpha2 and gamma band (for the complete

**Table 2. Mean CSD analysis.**

| Bands/ Condns. | Mean: Ms | Mean: SAR | Mean: BL | p-value: Ms Vs SAR | p-value: Ms Vs BL | p-value: SAR Vs BL |
|---|---|---|---|---|---|---|
| Alpha1 | 96.46941 | 28.12115 | 77.44804 | 0.03913 | 0.30848 | 0.01182 |
| Alpha2 | 97.13750 | 27.98740 | 78.65461 | 0.00074* | 0.024951 | 0.00558* |

Mean CSD comparison of the whole brain across Ms: sad music, SAR, and BL: baseline resting state condition.

co-ordinates of ROI, see S2 Table). We performed a mean brain activity comparison. A two-way repeated ANOVA with factor bands (3: alpha1, alpha2, and gamma) and conditions (3: baseline, SAR, and sad music listening) was administered. We found a significant interaction effect with a Greenhouse-Geisser correction between bands and conditions ($F_{1.698,23.767}$ = 5.370, p = 0.015). One-way repeated measure ANOVA was performed to study the simple effect of bands. Results showed a significant simple effect of bands with a Greenhouse-Geisser correction on brain activity for alpha1 ($F_{1.237,17.322}$ = 4.921, p = 0.034) and alpha2 ($F_{1.331,18.636}$ = 13.894, p = 0.001)). Post hoc analysis using Bonferroni correction revealed a significant difference between the Music listening condition and SAR condition in the alpha2 band only. Findings showed a significant enhancement in the mean CSD for the Music condition as compared to the SAR condition (t = 4.294, df = 14, p < 0.001, effect size = 1.1087). We also obtained a significant reduction in the mean CSD for SAR state as compared to the baseline in the alpha2 band (t = -3.2705, df = 14, p< 0.01, effect size = 0.8444). Table 2 shows the mean value of CSD (across 689 voxels) for different conditions (sad music, SAR, baseline) along with the p-value for the difference between the conditions.

## CSD analysis for sad music with SAR condition and baseline

CSD analysis of sad music and SAR conditions (across 689 voxels) showed significant variation in the alpha2 and gamma bands. Fig 3(a) shows a significant enhancement in the CSD in the alpha2 band during passive listening of sad music as compared to the SAR condition with t-value ranging from 3.9700 to 5.4927 and effect size ranging from 1.0251 to 1.4182. Most of the regions were located in the brain's central posterior (CC, PCC, PHC) part. Fig 3(b) depicts that most of the regions with enhanced CSD were located in the brain's right hemisphere compared to the left hemisphere. The difference was statistically significant (t = -4.3752, df = 14, p<0.01, effect size = -1.1297). Fig 3(c) shows a significant decrease in the CSD in the gamma band during passive listening of sad music as compared to the SAR condition with t-value ranging from -3.9935 to -6.4373 and effect size ranging from -1.0311 to -1.6621. Most of the regions were located in PCC. Fig 3(d) depicts that most of the regions with reduced CSD were located in the brain's right hemisphere compared to the left hemisphere. The difference was statistically significant (t = -31.1340, df = 14, p<0.001, effect size = -8.0388). Fig 3(e) shows a significant enhancement in the CSD in the alpha2 band during passive listening of sad music as compared to the baseline condition with t-value ranging from 3.6114 to 4.7791 and effect size ranging from 0.9325 to 1.234. Most of the regions were located in CC. Fig 3(f) depicts all the regions with enhanced CSD were located in the left hemisphere of the brain (t = 13.5695, df = 14, p<0.001, effect size = 3.5036).

## CSD analysis for SAR condition and baseline state

CSD analysis of SAR conditions and baseline showed significant variation in the alpha2 and gamma bands. Fig 4(a) shows a significant decrease in the CSD in the alpha2 band during SAR condition with t-value ranging from -4.0654 to -5.7798 and an effect size ranging from -1.0497

**Current Density Comparison**

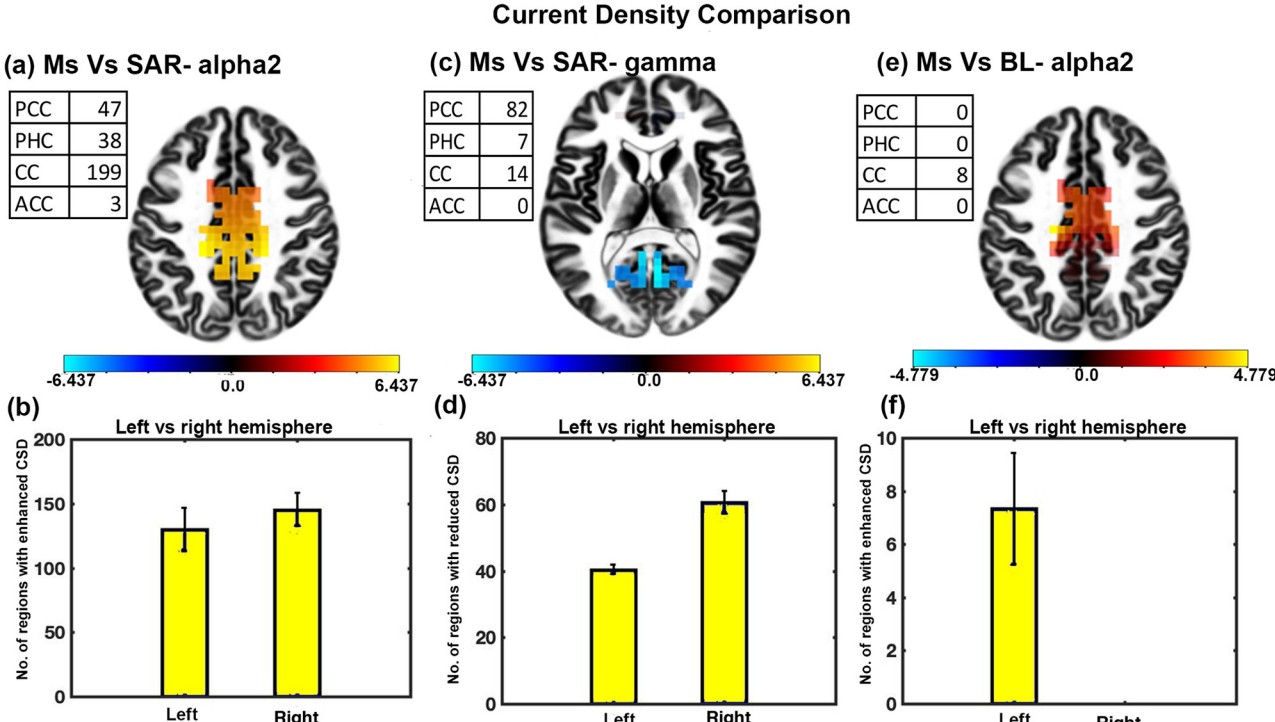

**Fig 3. CSD analysis for sad music with SAR condition and baseline.** (a) Slice frontal view representation of the brain regions depicting the change in the CSD in the alpha2 band (b) The hemispheric difference concerning the number of brain regions having significantly enhanced CSD during sad music intervention compared to SAR state. (c) Slice frontal view representation of the brain regions depicting the change in the CSD in the gamma band (d) Brain hemisphere lateralization with respect to the number of regions with decreased CSD in the gamma band (e) The Slice frontal view representation of the brain regions depicting the change in the CSD in the alpha2 band for sad music listening compared to baseline (f) Brain hemisphere lateralization with respect to the number of regions with increase CSD in alpha2 band (error bars = 1 SD).

to -1.4923. Most of the regions were located in the CC. Fig 4(b) depicts that most of the regions with reduced CSD were located in the brain's right hemisphere compared to the left hemisphere. The difference was statistically significant (t = -24.4616, df = 14, p<0.001, effect size = -6.3160). Fig 4(c) shows a significant enhancement in the CSD in the gamma band during the SAR condition with t-value ranging from 4.0697 to 6.1474 and an effect size ranging from 1.0508 to 1.5873. Most of the regions were located in the posterior (PCC, PHC) part of the brain. Fig 4(d) depicts that most of the regions with increased CSD were located in the brain's right hemisphere compared to the left hemisphere. The difference was statistically significant (t = -245.9086, df = 14, p<0.001, effect size = -63.4933).

## Subjective assessment of coping due to sad music

Subjective ratings of the phenomenological properties of the memories recalled during SAR were vividness (Mean = 4.2, SD = 0.67), reliving (Mean = 4.13, SD = 0.74), and age of the memory in months (Mean = 14.2, SD = 10.3). Subjective assessment of mood showed that participants experienced a larger negative state under SAR condition (Mean = 3.9, SD = 0.7) and sad music listening (Mean = 3.9, SD = 1.3) as compared to the baseline state (Mean = 0.4, SD = 1.9). The difference was significant for SAR condition (t = -8.663, df = 14, p<0.001, effect size = -2.236) as well as sad music listening (t = -6.094, df = 14, p<0.001, effect size = -1.5735) as shown in Fig 5 (a). However, results also showed no statistical difference in the experience

**Current Density Comparison**

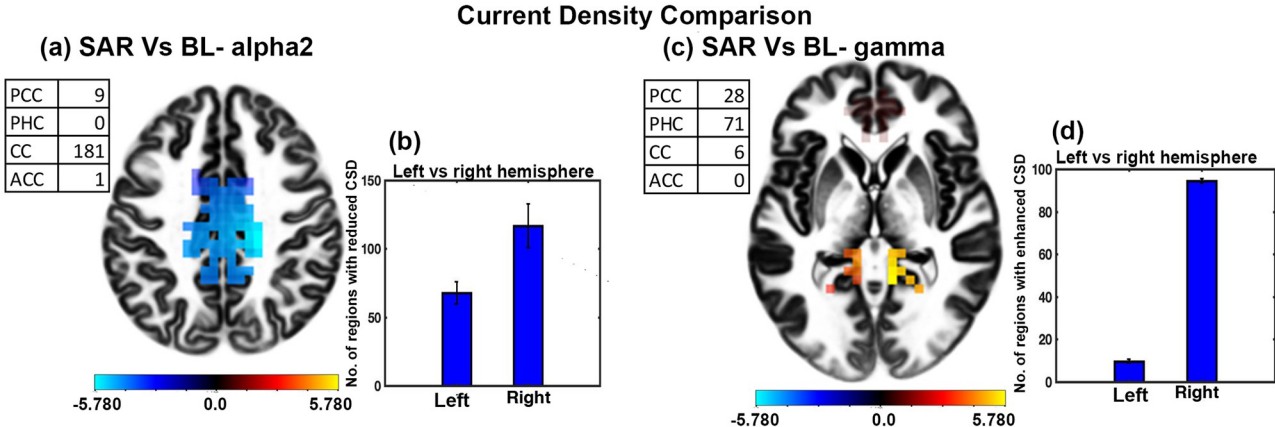

**Fig 4. CSD analysis for SAR condition and baseline state.** (a) Slice frontal view representation of the brain regions depicting the change in the CSD in the alpha2 band (b) The hemispheric difference concerning the number of brain regions having significantly reduced CSD under SAR condition compared to the baseline state. (c) Slice frontal view representation of the brain regions depicting the change in the CSD in the gamma band (d) The hemispheric difference concerning the number of brain regions having significantly enhanced CSD under SAR condition compared to the baseline state. (error bars = 1 SD).

under sad music listening and SAR condition (t = 0, df = 14, p<0.001, effect size = 0.5940). Fig 5(b) shows that re-experience effect (Mean = 3.7917, SD = 0.7858) was the most prominent self-regulatory goal achieved during sad music listening. It was higher than memory (Mean = 3.333, SD = 0.8772), distraction(Mean = 2.7778, SD = 1.0209), cognition (Mean = 3.0444, SD = 0.9666), and friendship (Mean = 3.2333, SD = 0.6974). However, the differences were not significant. Participants' responses also showed unanimous positive experiences after listening to the music (mean = 3.733, SD = 0.7037). Fig 5(c) illustrates the findings.

## Discussion

The current work examines the neural correlates and their alteration through lag phase synchronization and CSD analysis while the individuals passively hear sad music post adverse circumstances. We performed EEG source level analysis of the brain through eLoreta during (1) the brain's resting state, (2) an induced adverse state, and (3) intervention of a sad musical

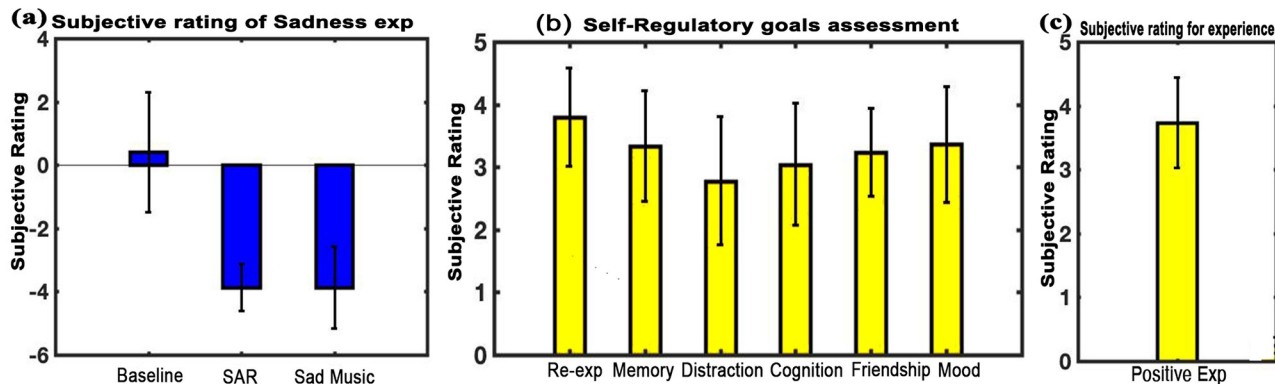

**Fig 5. Subjective assessment.** (a) Illustrates the mean subjective rating of experience across the three conditions of baseline, SAR, and sad music listening. (b) Illustrates the subjective rating for Self-Regulatory goal upon music listening (c) Illustrates the mean subjective assessment for sad music. Results show a positive experience of listening to music. (error bars = 1 SD).

excerpt. Our ROI are emotion and memory processing regions, specifically ACC, PCC, CC, and PHC.

Our first analysis shows that mean lag phase synchronization is significantly enhanced during sad music listening with respect to the SAR state in the ROI for the alpha band (Table 1). Lag phase synchronization between two regions measures interareal connectivity between them [71]. This implies an increase in the mean interareal connectivity within the ROI in the alpha band during sad music listening. Klimesch *et al.* [72] investigated the nature of phase synchronization (interareal) in the alpha band and found an increased phase synchronization among the task-relevant regions in processing stimulus/task. Several studies have further reported that alpha phase synchronization among task-relevant regions mediates high task performance ranging from internal mental calculation [73] to creativity [74], to visual-spatial attention [75], and motor activity [76]. The review conducted by *Palva et.al.* [71] concludes that alpha phase synchronization among the task-relevant regions is directly proportional to the cortical activity of the regions and postulates an "active-processing hypothesis" for alpha band phase synchronization. Additionally, the alpha band has been consistently shown to be involved in semantic memory processing and represents the content of episodic memory [45].

Further functional connectivity analysis through eLoreata show that most interconnections in alpha1 and alpha 2 bands are present between the spatial processing area (PCC) and memory related area (PHC) during sad music listening state compared to the SAR state (Fig 1). In addition, findings also show enhanced connectivity between memory-related area (PHC) and emotion-related area (ACC) in the alpha1 band. Our results are in line with earlier functional connectivity studies on music [52–54] and signify an increased content-specific information processing primarily, the spatial context of the episode represented through PCC and PHC connectivity [32].

Results also show right hemisphere lateralization for the alpha2 band as expected, in line with earlier findings of right hemisphere dominants for negative memories [40–44]. However, we obtained left hemisphere lateralization for the alpha1 band. This reverse nature of lateralization for alpha1 and alpha2 was also reported in an earlier study [77].

Resting state functional connectivity, rather than merely reflecting passive or epiphenomenal activity, is associated with active processing of high domain cognition [78] in line with the active-processing hypothesis [71]. A review by *Stevens et.al.,* [78] demonstrated a positive correlation between the strength of resting state functional connectivity to behavioral and cognitive domains, including perception, language, learning and memory. Mean lag phase synchronization analysis between SAR state and baseline resting state reveals a significant reduction in the alpha1 and alpha2 bands in ROI for SAR state compared to baseline resting state (Table 1). Further analysis reveals that most interconnections with reduced connectivity are among areas connected to spatial processing (PCC), memory related areas (PHC), specifically in the alpha2 band (Fig 2). This reduction implies decreased content-specific information processing during SAR state compared to the baseline resting state. The finding supports an earlier finding that observed a reduction in the coherence of the various elements in a negative episodic memory experiment [79].

We did not observe any significant difference in the functional connectivity between the sad music listening condition and baseline resting state for the alpha band (Table 1), as it was observed in a direct comparison between the two state in earlier studies [52–54]. This may be due to a prolonged sad autobiographical period of diminished phase synchronization between baseline resting state and sad music listening. Future studies with a musical stimulus of larger durations than the SAR period are needed to ascertain it.

We also observed a significant reduction of mean lag phase synchronization of the brain, implying diminished mean inter-connectivity in the gamma band during sad music listening

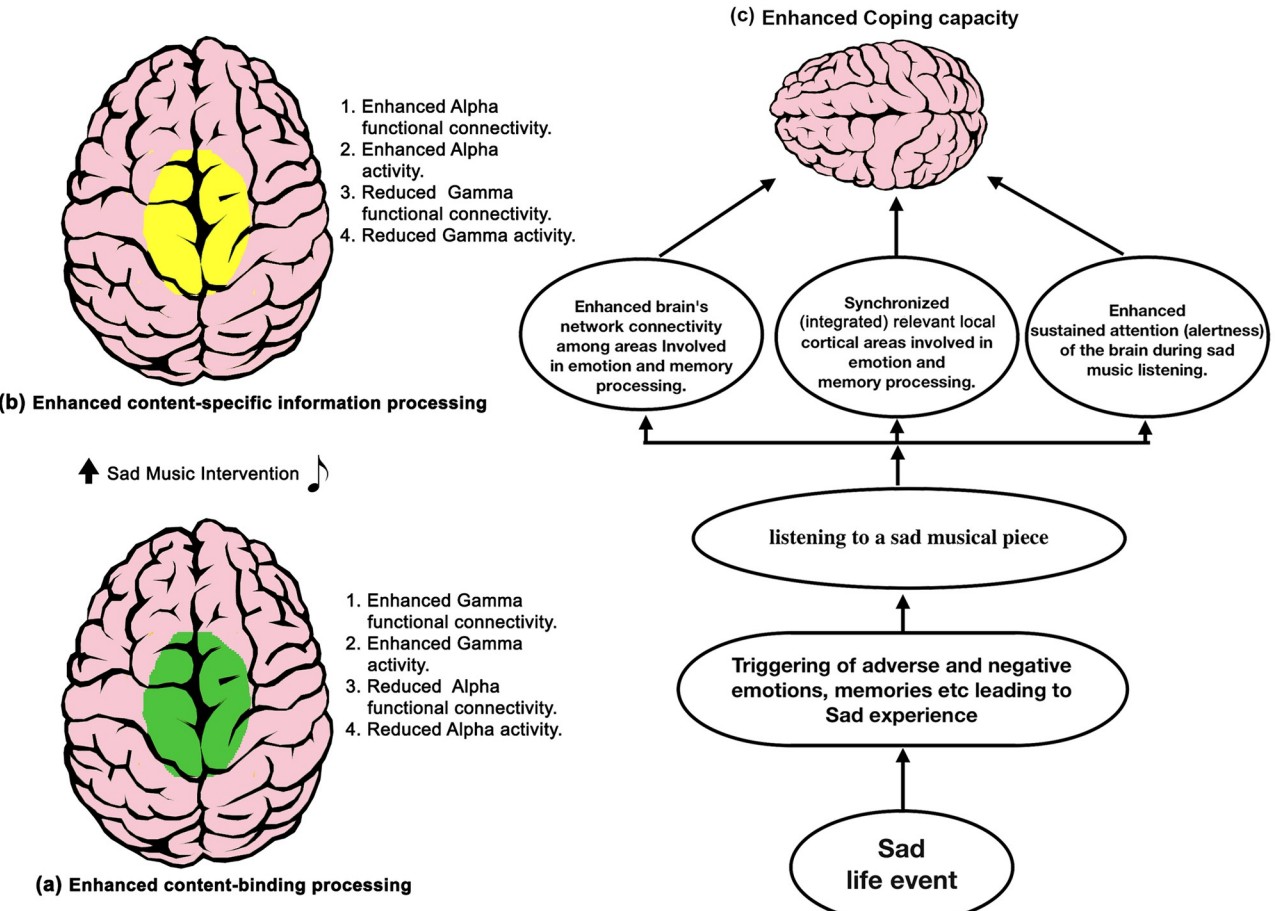

**Fig 6. Schematic model.** (a) Brain under SAR is characterized by enhanced functional connectivity and activity in gamma band, (Green colour) (b) Brain under sad music listening is characterized with enhanced coherence, and activity in the alpha band (yellow colour). (c) Potential neural mechanisms that boost up the coping ability in adverse circumstances are (1) enhancing functional connectivity of the brain, (2) enhancing local cortical integration (synchronization) of relevant areas, and (3) enhancing sustained attention (alertness) of the brain.

with respect to the SAR state in ROI (Table 1). The Gamma band has been consistently shown to be related to episodic memory [45]. The Gamma band plays a vital role in binding the contents received from several different regions of the brain into an episode [45]. Further eLoreta analysis shows significantly enhanced lag-phase synchronization for several connections in the gamma band during the SAR state compared to sad music listening (Fig 1) and baseline resting state (Fig 2). Findings show most of the interconnections were among spatial processing area (PCC) and memory related area (PHC). The enhanced gamma band's functional connectivity during SAR state is expected and validates episodic retrieval by the subjects during SAR.

Although we observe the difference in mean lag-phase synchronization analysis in the theta and Beta2 bands but the difference did not reach a significant level under Bonferroni correction.

Our second investigating tool was the measurement of brain activation estimated through CSD. CSD at a given brain region measures local cell assembly synchronization of the region [71]. Results show significantly reduced mean CSD activity in ROI during SAR state compared to baseline and upon sad music listening in the alpha2 band (Table 2). Detailed CSD analysis between SAR condition and baseline resting state shows reductions in the alpha2 activity

during SAR state (Fig 4). Most regions with reduced CSD were CC and PC with right hemisphere lateralization. SAR state is an externally directed activity characterized by ERD. ERD studies have shown a negative correlation between brain activity with the alpha CSD activity of the relevant cortical area [80] Our results are expected as per the earlier studies [45, 81, 82] and are in line with cortical inhibition hypothesis [72].

Analysis also showed significantly enhanced alpha2 CSD activity during sad music listening compared to the baseline resting state and SAR state (Fig 3). All the significantly enhanced regions during sad music listening compared to the baseline resting state were located around the central-posterior part of the brain. Listening to music has been linked to increased internal attention, alertness and diminished external attention [48, 49]. The state is characterized by an increased ERS, mainly around the central-parietal area in the alpha band. It signifies local integration of relevant cortical areas and inhibition of irrelevant cortical areas [48, 49]. Our results are in line with the earlier studies [48–51] and suggest a state of enhanced internal directed activity (alertness and emotion processing) during sad music listening as compared to baseline.

Whole ROI had significantly enhanced CSD during sad music listening when compared to SAR state. Regions were primarily located in the right hemisphere. Results are well in line with earlier studies depicting the difference between external and internal directed activity [83, 84]. and shows a clear enhanced alpha activity during internally directed activity [48, 49, 83, 84]. Thus, our results suggest enhanced internal processing and alertness during sad music listening as compared to SAR state [34–36, 50, 85].

Detailed CSD activity analysis in the gamma band shows significantly enhanced activity during SAR state compared to sad music listening and baseline resting state especially in PCC, PHC, and CC. Gamma activity is a robust signature of episodic memory. Enhanced gamma activity during SAR state is expected as participants recall a sad episode from their life. These results are in line with the existing studies [47, 86].

Subjective assessment of participants' mood across the three conditions shows that the participants experienced an enhanced sad state during SAR state and sad music listening compared to the baseline resting state. Although, participants' experience was sad during SAR state as well as during sad music listening. There seems to be a qualitative difference in the nature of sadness as reflected through the self-regulatory questionnaire. The questionnaire reveals that the sad musical excerpts facilitated the participants to achieve several self-regulatory goals [3, 16] (Fig 5). Especially re-experience effect, mood enhancement and memory. Finding also shows that listening to experimenter-selected sad music post-SAR gave them an overall positive experience, in line with earlier studies [7, 57].

The findings show that the brain during the listening of sad music compared to SAR is characterized by enhanced functional connectivity in the alpha band, which implies enhanced information processing during sad music listening. This enhanced processing could explain the self-regulatory goals achieved during sad music listening compared to adverse circumstances as frequently observed in earlier studies [3, 5, 16]. A key finding of this study is enhanced connectivity between and within PHC and PCC in the alpha bands during sad music listening compared to SAR state. This possibly explains the enhanced memory retrieval observed in earlier [3] as well as the current study. The results support alpha band's "active-processing hypothesis" [71, 87].

A review by *Kiyonaga et.al* shows that internal and external processes share a common cognitive resource and thus compete with each other. An increase in externally oriented activity comes at the cost of reduced internal processing and vice versa [88]. Sad music listening state, in comparison to SAR state, is characterized by the absence of externally directed activity (writing) and increased internal alertness due to music listening [48, 49], thus facilitating

enhanced internal processing and internal alertness [48, 49, 88]. We observed increased activity (local cortical synchronization) at all the places PCC, PHC, CC and ACC in our ROI during sad music listening. The result suggests enhanced internal alertness and internally directed processing during sad music listening compared to SAR state and baseline. The enhanced internal processing directly and internal alertness indirectly may also facilitate the self-regulatory goals (improved internally directed process like self-referential emotion and memory processing) achieved during music listening post adverse circumstances.

Gamma band analysis between sad music listening and SAR state shows enhanced functional connectivity and brain activity for SAR state. Gamma band specifically is involved in binding the various contents of the event into a cohesive episode, as discussed earlier. Thus, the results suggest that the brain during sad music listening is better in content processing (estimated through alpha band analysis). During SAR state, it is better in cohesive episodes formation. Brain analysis for sad music listening and baseline resting state reveals that the states differ only in brain activity (CSD) in the alpha2 band. It suggests an enhanced alert state during music listening. Brain analysis between SAR state and baseline resting state reveals that SAR state is characterized with enhanced episodic recollection (estimated through gamma band analysis) as expected.

Thus, findings suggest that in a healthy population, listening to sad music post adverse circumstances is characterized by enhanced content (emotions and memories) processing via three possible neural mechanisms in the alpha band (Fig 6):- (1) by enhancing the brain's network connectivity among areas involved in emotion and memory processing, (2) by enhancing local cortical integration (synchronization) of relevant areas involved in emotion and memory processing, and (3) by enhancing sustained attention (alertness) of the brain during sad music listening.

Our results in summary clearly show that the brain's neural correlates during sad music listening are unique and distinct from the SAR state and the baseline. Thus, it supports the view that the negative state induced during music is real but distinct from the SAR state and the benefits of sad music listening are not just an artifacts of the aesthetic nature of music.

However, the current model needs to be tested with different kinds of sad music, and mood inductions procedures. The present study involved only male participants. Studies suggest dissimilar networks being used by males and females for processing emotion [55], thus limiting the generalizability of the findings. We need to check whether listening to sad music can further enhance the internally oriented process when the mood induction procedure is similar (internally oriented). The internally oriented mood induction method would help to investigate wider brain regions. Sad music is used as a coping strategy across several cultures. The present results give a promising hypothesis to validate and extend the findings to other cultures and practices. We wonder whether greater left-sided symmetry observed during enhanced functional connectivity in the alpha1 band is associated with the positive/approach emotion aspect [89] of listening to sad music compared to SAR state. Further studies are required to ascertain it. The Salience network plays a vital role in social and affective tasks involving emotions and introspections [90]. The Salience network compromises of the anterior insula, dorsal anterior cingulate cortex, and three key subcortical structures: amygdala, ventral striatum, and substantia nigra/ventral tegmental area [91]. Our results do support the benefit of listening to music in terms of emotion regulation and enhanced alertness. However, although EEG analysis is favoured with the high temporal resolution, nevertheless, it lacks spatial resolution. Future studies through fMRI would give better spatial resolution and expand the research by quantifying the involvement of cortical and sub-cortical regions viz salience network in emotional coping through sad music listening. Although neural correlates difference between sad music listening and SAR gives potential mechanisms responsible for the self-

regulatory goals, further studies are needed to establish the causal connections between the observed effect and sad music excerpt.

## Supporting information

**S1 Table. Co-ordinates for the lag Phase synchronization analysis.** 27 ROI defined according to the standard Montreal Neurological Institute (MNI) template and Brodmann areas (BA) in the area of our analysis.
(DOCX)

**S2 Table. Co-ordinates for the CSD analysis.** 689 Voxels are defined according to the standard Montreal Neurological Institute (MNI) template in our analysis area.
(DOCX)

**S3 Table. Power analysis.** Statistical strength analysis of the experiment.
(DOCX)

## Acknowledgments

The authors are thankful to Mr. Chandan Kumar and Saurav Dutta for their help in data collection and figure preparation. This work is a part of A.G.'s doctoral thesis. The authors are especially grateful to the judges, namely Mrs Brijbala Narain, Mrs Kakoli sarkar, Mrs Meena gupta, Mr K K sarkar and Mrs Deepti Gautam.

## Author Contributions

**Conceptualization:** Ashish Gupta, Braj Bhushan, Laxmidhar Behera.

**Data curation:** Ashish Gupta.

**Formal analysis:** Ashish Gupta.

**Funding acquisition:** Laxmidhar Behera.

**Investigation:** Ashish Gupta.

**Methodology:** Braj Bhushan.

**Project administration:** Laxmidhar Behera.

**Resources:** Ashish Gupta.

**Software:** Ashish Gupta.

**Supervision:** Laxmidhar Behera.

**Validation:** Ashish Gupta.

**Visualization:** Braj Bhushan.

**Writing – original draft:** Ashish Gupta.

**Writing – review & editing:** Braj Bhushan, Laxmidhar Behera.

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
