## [Decision Letter · Decision Letter 0]

15 Sep 2022

PONE-D-21-33402Neural response to Sad Autobiographical recall and Sad Music listening post recall reveals distinct brain activation in alpha and gamma bands.PLOS ONE

Dear Dr. Behera,

Thank you for submitting your manuscript to PLOS ONE. After careful consideration, we feel that it has merit but does not fully meet PLOS ONE’s publication criteria as it currently stands. Therefore, we invite you to submit a revised version of the manuscript that addresses the points raised during the review process.

Both Reviewers' comments converge on a significant revision, including methodological issues that the authors must address. I fully agree with their decisions. I invite the authors to revise their manuscript and resubmit it for further consideration.

We look forward to receiving your revised manuscript.

Kind regards,

Vilfredo De Pascalis

Academic Editor

PLOS ONE

Journal Requirements:

2. PLOS ONE does not copy edit accepted manuscripts (https://journals.plos.org/plosone/s/criteria-for-publication#loc-5). To that effect, please ensure that your submission is free of typos and grammatical errors.

Additional Editor Comments (if provided):

Both Reviewers' comments converge on a significant revision, including methodological issues that the authors must address. I fully agree with their decisions. I invite the authors to revise their manuscript and resubmit it for further consideration.

Reviewers' comments:

Reviewer's Responses to Questions

**Comments to the Author**

1. Is the manuscript technically sound, and do the data support the conclusions?

Reviewer #1: Yes

Reviewer #2: No

2. Has the statistical analysis been performed appropriately and rigorously? 

Reviewer #1: Yes

Reviewer #2: Yes

3. Have the authors made all data underlying the findings in their manuscript fully available?

Reviewer #1: Yes

Reviewer #2: Yes

4. Is the manuscript presented in an intelligible fashion and written in standard English?

Reviewer #1: No

Reviewer #2: Yes

5. Review Comments to the Author

Reviewer #1: Review for PONE_21-33402

Title: Neural response to Sad Autobiographical recall and Sad Music listening post recall

reveals distinct brain activation in alpha and gamma bands.

General Overview:

Summarise main findings:

Overall, this is an innovative study, which was conceptualised, designed, and will certainly add to the literature. The study explored the EEG source strength, connectivity and changes associated with sad autobiographical recall and listening to sad music. In particular, alpha band lag phase-synchronization during sad music listening, especially within and between the Posterior cingulate cortex (PCC) and (PHC) when compared to sad autobiographical recall. The main feature found was that there may be two distinct for sad music listening and sad recall. The effects were found in alpha and gamma band.

Limitations and Strengths:

Overall, this is an innovative study, and well conceived. However there may be a few minor issues requiring clarification and restructuring of the narrative. For example:

• Why not other EEG frequencies, like theta? Beta? They are discussed I the introduction in the context of emotional memory. The authors need to explain why significance was not seen for these bands.

• Laterality could be associated with positive and negative emotion (Ahern et al 1985). Some disorders like PTSD may have triggers associated with sad music; perhaps the authors may need to highlight this early.

• The research may be culturally sensitive and will need further research in other cultures.

• Perhaps suggest reorganising format so Methods appears before results. There are some explanations in Methods which assist with the interpretation of the Results.

• Why not also have an additional 20 female participants?

The study is unique and will certainly add to the literature and be hypothesis generating especially for other clinical and cultural scenarios.

Abstract

• A good summary of the conceptual framework, methodologies, results, and outcomes. Could be more concise and clear.

Introduction

• Concepts are generally well introduced. Good rationale presented early in the narrative.

• However, the first three paragraphs could be more concise, seems like the same concepts are being covered again. Need to make it clear from a behavioral, social and biological point of view.

• The research may be culturally sensitive and will need further research in other cultures. Is it relevant to other cultures?

• Should also address emotional regulation in the context of the salience network as well.

• The final paragraph clarifies the rationale and outcomes , however the term ”probing tool” may not be an accurate description for the methodologies used. Reference to beta but no data presented in Figures later.

• May require some restructuring to improve flow of the narrative and rationale.

• Overall well referenced and conceptual design supported.

Results

• Well-constructed overall

• Some of the methodology described in Results, should be covered in Methods, for example, little mention of functional coherence analysis in Methods.

• Line 152-153 for the first time indicates that only the 3 bands reported significance across conditions. It may be advisable to indicate this in Methods as well.

• Need to refer to Tables in narrative eg (See Table 1).

Discussion

• A well-constructed narrative, however some proof reading and restructure necessary to improve flow of the narrative would be useful.

• Avoid the use of the term “probing tool”, investigative tool may be more appropriate.

• Some large paragraphs may require some rewriting to improve flow of the narrative. Starting line 283

• Some repetition

• Line 398 “In summary..”, should be a short paragraph and not followed by new and additional narrative. “In summary..”, again at line 459

• A limitations narrative is very brief and could have been expanded.

Methods

• This section is well constructed overall; however subheadings may have been useful in clarifying the protocols etc

• An appropriate methodology for functional connectivity was selected and applied. However more information about why, is required.

• Why do authors cite eLORETA paper ? Did they use this technique? Or was it sLORETA alone? This needs to be clarified. eLORETA is not the same as sLORETA and there may be some confusion in interpretation of data.

• Eye, blink and muscle movement may have contributed to EEG changes during SAR. How was this controlled for and was the minimize movement enough? Could this be achieved post analysis, not just visual inspection? Should reference the algorithms used in EEGLAB toolbox.

• What was the final number of participants tested once some were removed for previous exposure to music, artifact and after the KPD scale?

• Brand of amplifier/supplier ref?

• Why was an earlobe reference used? What about linked earlobes? This would contribute to laterality shifts.

• Was the baseline resting state eyes open or closed?

• Do the authors find any relationships with other bands, seeing that had these data.

• More details about how the sLORETA data was calculated and statistically analysed.

• A power analysis may have been useful in highlighting the statistical strength of effects and sLORETA data. Is n <20 sufficient for significance?

• Some of the methodology described in Results, should be covered in Methods, for example, little mention of functional coherence analysis in Methods.

• What platform was used to calculate the statistics? SPSS, Toolbox?

• Why was phase coherence selected as a tool?

References:

• Excellent review of the field with most primary references used.

Figure Legends:

• Most images are well constructed and easy to follow.

Reviewer #2: Manuscript ID: PONE-D-21-33402

Title: Neural response to sad autobiographical recall and sad music listening post recall reveals distinct brain activation in alpha and gamma bands.

Authors: Ashish Gupta, Braj Bhushan and Laxmidhar Behera

In this research, Gupta and colleagues compared EEG rhythms collected at rest (baseline) with those induced by sad autobiographical recall (SAR) and upon exposure to sad music. Results focused in particular on sad music listening condition, and revealed a significant involvement of Alpha and Gamma EEG bands in specific region of interest (ROIs) including CC, ACC, PCC and PHC. Authors concludes that the SAR condition induced enhanced gamma-band activity, suggesting increased content binding capacity, whereas the sad music listening was associated with an enhanced alpha band activity, suggesting increased content-specific information processing.

The paper addresses an important topic. However, there are important issues in the methods and data discussion that significantly affect the quality of the manuscript.

First of all, I’m puzzled about the authors’ choice over the frequency range used to define the delta EEG rhythm (i.e., 1.5-6 Hz): usually, delta waves range between 0/0.5/1 Hz to 4 Hz, whereas 4-6 Hz represent the lower frequencies of theta rhythm (full range = 4-8 Hz). I recommend to revise the analyses on delta and theta bands adjusting the frequency ranges, in line with most of qEEG literature.

The second, critical issue regards results discussion and interpretation. All the discussion focused on the increased phase synchronization/connectivity in the Alpha (1 and 2) band during sad music listening vs. SAR condition, and authors interpreted these findings as “an overall enhancement in the brain’s processing in areas related to memory and spatial/action information region specifically, in addition to emotion”. I’m not convinced this is the correct interpretation, as alpha rhythm is physiologically associated with cortical inhibition (see, for example, Klimesch, W. (2012). Alpha-band oscillations, attention, and controlled access to stored information. Trends in Cognitive Sciences 16, 606-617). In this perspective, increased alpha connectivity should be interpreted as increased inhibition of these regions in the brain during sad music listening, or, from the other perspective, a decreased inhibition of the memory/spatial-action information/emotion circuit during SAR condition. This interpretation is further supported by the lack of “any significant difference between the functional connectivity between sad music listening condition and baseline resting-state for alpha band”.

Whether sad music listening induces “an overall enhancement in the brain’s processing in areas related to memory and spatial/action information region specifically, in addition to emotion”, this may be found also with respect to resting state (baseline) condition.

In addition, the decreased connectivity in alpha band (suggesting decreased inhibition within the brain network) found in SAR vs. sad music listening condition matches the result found in gamma EEG band, as “[…] Detailed brain activity analysis in the gamma band shows significantly enhanced activity during SAR state compared to sad music listening AND BASELINE RESTING STATE […] Gamma activity is a robust signature of episodic memory.”

I therefore recommend that the authors reconsider their interpretations of alpha rhythm in a more coherent discussion, in line with the suggested evidence.

Minor: was the participants’ subjective assessment of mood - across the three conditions - significantly correlated with EEG alpha and gamma connectivity patterns?

6. PLOS authors have the option to publish the peer review history of their article (what does this mean?). If published, this will include your full peer review and any attached files.

Reviewer #1: **Yes: **Joseph Ciorciari

Reviewer #2: No

---

## [Author Response · Author response to Decision Letter 0]

1 Dec 2022

Reviewer 1: Thank you for the valuable comments, feedback, and recommendations, which offered us an opportunity to improve the quality of the manuscript. We have sincerely taken into account all the comments and have tried our best in revising the manuscript accordingly. They are very helpful. We have responded to all the queries and comments. 

Reviewer 2: Thank you for the valuable comments, feedback, and recommendations, which offered us an opportunity to improve the quality of the manuscript. We have sincerely taken into account all the comments and have tried our best in revising the manuscript accordingly. They are very helpful. We have responded to all the queries and comments.

---

## [Decision Letter · Decision Letter 1]

15 Dec 2022

Neural response to Sad Autobiographical recall and Sad Music listening post recall reveals distinct brain activation in alpha and gamma bands.

PONE-D-21-33402R1

Dear Dr. Behera,

We’re pleased to inform you that your manuscript has been judged scientifically suitable for publication and will be formally accepted for publication once it meets all outstanding technical requirements.

Kind regards,

Vilfredo De Pascalis

Academic Editor

PLOS ONE

Additional Editor Comments (optional):

I see that the manuscript has been revised in line with both Reviewers' comments. I thank the authors for addressing all comments made during the review.

The quality of the manuscript is now significantly improved. The study is now ready for publication.

Reviewers' comments:

Reviewer's Responses to Questions

**Comments to the Author**

1. If the authors have adequately addressed your comments raised in a previous round of review and you feel that this manuscript is now acceptable for publication, you may indicate that here to bypass the “Comments to the Author” section, enter your conflict of interest statement in the “Confidential to Editor” section, and submit your "Accept" recommendation.

Reviewer #1: All comments have been addressed

Reviewer #2: All comments have been addressed

2. Is the manuscript technically sound, and do the data support the conclusions?

Reviewer #1: Yes

Reviewer #2: Yes

3. Has the statistical analysis been performed appropriately and rigorously? 

Reviewer #1: Yes

Reviewer #2: Yes

4. Have the authors made all data underlying the findings in their manuscript fully available?

Reviewer #1: Yes

Reviewer #2: Yes

5. Is the manuscript presented in an intelligible fashion and written in standard English?

Reviewer #1: Yes

Reviewer #2: Yes

6. Review Comments to the Author

Reviewer #1: We thank the authors for commenting and addressing all comments made during the first review. The revised manuscript is well written and complete.

Reviewer #2: The manuscript has been revised according to my previous review comments, and I think it is now significantly improved and suitable for publication on PLoS ONE.

7. PLOS authors have the option to publish the peer review history of their article (what does this mean?). If published, this will include your full peer review and any attached files.

Reviewer #1: **Yes: **Associate Professor Joseph Ciorciari

Reviewer #2: No

---

## [Editor Report · Acceptance letter]

27 Dec 2022

PONE-D-21-33402R1 

Neural response to Sad Autobiographical recall and
Sad Music listening post recall reveals distinct brain
activation in alpha and gamma bands 

Dear Dr. Behera:

I'm pleased to inform you that your manuscript has been deemed suitable for publication in PLOS ONE. Congratulations! Your manuscript is now with our production department. 

Kind regards, 

on behalf of

Prof. Vilfredo De Pascalis 

Academic Editor

PLOS ONE